# Periovulatory Subphase of the Menstrual Cycle Is Marked by a Significant Decrease in Heart Rate Variability

**DOI:** 10.3390/biology12060785

**Published:** 2023-05-29

**Authors:** Ajna Hamidovic, John Davis, Margaret Wardle, Aamina Naveed, Fatimata Soumare

**Affiliations:** 1Department of Pharmacy, University of Illinois at Chicago, 833 S. Wood St., Chicago, IL 60612, USA; 2Department of Psychiatry, University of Illinois at Chicago, 1601 W. Taylor St., Chicago, IL 60612, USA; 3Department of Psychology, University of Illinois at Chicago, 1007 W. Harrison St., Chicago, IL 60607, USA

**Keywords:** heart rate variability, luteinizing hormone, menstrual cycle, ovulation, progesterone

## Abstract

**Simple Summary:**

If measured in the high-frequency (HF) domain, heart rate variability (HRV) reflects the tonic, inhibitory control by the parasympathetic nervous system (PNS) to decelerate the heart rhythm. Several studies have examined changes in HF-HRV variability across the menstrual cycle; however, they did not implement the latest technological and statistical approaches. The evaluation of HF-HRV across the menstrual cycle is important because low HF-HRV variability is observed in numerous conditions, including heart disease and depression. Moreover, if a change in HF-HRV is observed around ovulation, it may be used to time conception. This study found that HF-HRV significantly decreases around ovulation relative to all other times of the menstrual cycle. This finding enhances our understanding of human biology. Future studies may examine whether the decrease in HF-HRV is a normal process that may be utilized in the future to enhance fertility rates, or whether it represents a biological process that may predispose some women to disease development.

**Abstract:**

(1) Background: High-frequency heart rate variability (HF-HRV) is an essential ultradian rhythm that reflects the activity of the PNS to decelerate the heart. It is unknown how HF-HRV varies across the menstrual cycle (MC), and whether progesterone mediates this potential variation. (2) Methods: We enrolled 33 women in the study to attend eight clinic visits across the MC, during which we measured their resting HF-HRV and collected samples for the analysis of luteinizing hormone (LH) and progesterone. We realigned the study data according to the serum LH surge to the early follicular, mid-follicular, periovulatory, early luteal, mid-luteal and late luteal subphases. (3) Results: Pairwise comparisons between all the subphases showed significant differences between the early follicular and periovulatory subphases (β = 0.9302; *p* ≤ 0.001) and between the periovulatory and early luteal subphases (β = −0.6955; *p* ≤ 0.05). Progesterone was positively associated with HF-HRV in the early follicular subphase but not the periovulatory subphase (*p* ≤ 0.05). (4) Conclusions: The present study shows a significant drop in HF-HRV in the anticipation of ovulation. Further research in this area is critical given the marked cardiovascular disease mortality in women.

## 1. Introduction

Physiological rhythms are distinct time-related processes that are essential for the survival of the organism. They span several orders of magnitude, from microsecond to seasonal events [1,2]. Lasting more than 24 h, the ovulatory menstrual cycle is an infradian rhythm [3] during which basal body temperature, for example, increases by 0.4 °C on average compared to the follicular phase [4,5,6]. This and other rhythms synchronize cellular activities to enhance the stability of the organism, or alternatively, fail to do so, causing perturbations and disease progression [7].

Heart rate variability (HRV) is one such physiological rhythm that reflects the natural variation in the inter-beat interval between consecutive heartbeats. The autonomic nervous system dually innervates the heart. Enhanced sympathetic and parasympathetic tone increases and decreases heart oscillations, respectively. Frequency (HF-HRV; 0.15–0.40 Hz) and time (for example, the standard deviation of NN intervals (SDNN)) are measures of parasympathetic activity. The measurement of sympathetic activity using heart rate measurements (i.e., low-frequency HRV; LF-HRV) is controversial [8]. Hence, the present study focused on HF-HRV analyses, and when discussed in relation to the other literature (examining both time and frequency domain markers), it is referred to as cardiac vagal activity (CVA). 

Under resting conditions, as measured in the present study, high CVA echoes an increased parasympathetic (vagal) tone. Greater parasympathetic withdrawal (lower CVA) is reflective of poor adaptability and worse health [9]. Generally, the results of time and frequency methods are highly correlated, though not perfectly interchangeable. We carefully selected the moving polynomial method for quantifying HF-HRV in the present study for the following reasons: 1. It better reflects the contributions of the parasympathetic nervous system, our primary interest, compared to other methods, such as SDNN [10], 2. It shows superior statistical properties to those of standard time and frequency domain methods [11], and, 3. It does not require correction for respiration frequency, simplifying data collection [11,12].

Decades of theoretical and epidemiological research has produced an impressively long list of works outlining sophisticated CVA measurement techniques [13] and associations between low CVA and disease states [14]. However, whether and how CVA varies across the entire menstrual cycle is not clear due to the complexity of studying the cycle. The duration and ovulation timings vary both within and between women [15,16], and the underlying hormonal shifts, reflecting different menstrual cycle subphases, are unknown at the time of clinic visits. Hence, data collected across the menstrual cycle is inherently misaligned. A validated method to deal with this complexity is to schedule clinic visits at the estimated early follicular, mid-follicular, periovulatory (three consecutive daily clinic visits to optimize the capturing of the peak serum LH surge) upon which the data are realigned *post hoc*) [17]. This method, developed in the Biocycle study, was shown to outperform the “fixed cycle” method (all women come for a blood draw visit on days 13–15), the “backward” counting method (assuming the luteal phase averages approximately 14 days) and the “midpoint method” (assuming that the LH surge occurs around the midpoint of the cycle, and therefore, the 3-day window is created around the midpoint of the self-reported cycle length).

Assessments of CVA in relation to the menstrual cycle have been completed implementing a variety of approaches in an attempt to capture its potential changes, and the majority have not considered LH variation. For example, a recent meta-analysis [18] included studies that attempted to time menstrual changes in CVA according to key hormonal events based on self-reported cycle length, cycle day or the assumption that the luteal phase is 14 days [19,20]. Underlying assumptions of these methods, however, are that women can reliably report their cycle length and/or that the timing of hormonal fluctuations is the same across women, which is not the case [21,22,23]. The meta-analysis implemented the forward- and the backward-cycle-based *phase* determination to contrast follicular vs. luteal resting CVA and an ovulation-based *sub*phase determination to contrast CVA according to subphases. For the ovulation-based subphase determination, none of the studies implemented a serum measure of LH peak, which is a validated method to reclassify menstrual cycle subphases post hoc [17], or implemented a daily collection of first-morning urine specimens, which is the ‘‘gold standard’, as it ensures that critical hormone windows are accurately captured. However, the protocol is burdensome and may significantly affect recruitment and compliance. Instead, the studies included in the meta-analysis [18] defined ovulation based on (1) the day of the menstrual cycle [24,25,26,27,28], (2) basal body temperature [29] or (3) an ovulation prediction kit positive test result [30,31]. Otherwise, in the *phase* determination method in the meta-analysis [18], progesterone and/or estradiol of the included studies were sometimes measured at arbitrary timepoints, which may be inadequate given sex hormone variability within and across women [32]. As none of these studies implemented a validated method for menstrual subphase classification, variation in CVA across the entire menstrual cycle remains uncertain.

Following the publication of the meta-analysis by Schmalenberger et al. [18], additional groups examined CVA changes across the menstrual cycle. Sims et al. [33] demonstrated a gradual decline in CVA across the menstrual cycle divided into four equally spaced intervals: early follicular, late follicular, early luteal and late luteal. They found that CVA was significantly higher in the menstrual phase relative to the premenstrual phase when CVA was measured within cycle days 1–3 and within 5 days prior to the next cycle [34]. However, Ramesh et al. [35] found no difference in baseline CVA according to serum-verified low-estrogen (cycle day 1) and high-estrogen (cycle day 14) menstrual phases in premenopausal women. In a subsequent study by Schmalenberger et al. [31], CVA was lower in the mid-luteal phase relative to the perimenstrual, mid-follicular and ovulatory subphases based on a combination of forward and backward cycle counting in participants blinded to the menstrual cycle focus, and relative to the perimenstrual and ovulatory subphases based on positive ovulation test results.

In an attempt to examine the utility of a small, wireless sensor worn on a finger that captures physiological changes, including those of heart rate, Grant et al. [36] recently examined sleeping CVA as a predictor of LH surges. Urinary estradiol and progesterone metabolites surrounding LH surges were measured daily (i.e., “the gold standard”, as discussed above). Their analysis revealed a consistent pattern of HRV ultradian rhythm (2–5 h) power decrease; specifically, the ultradian HRV (RMSSD) power displayed a sharp trough an average of 2.11 (±1.27) days after LH surge onset. Whereas this study efficiently examined CVA around the time of ovulation, our study adds to this understanding by examining CVA across the entire menstrual cycle.

We first implemented the validated Biocycle study method to align HF-HRV data across study participants to the same biological window (i.e., subphase). Next, we conducted pairwise comparisons of all six subphases of the menstrual cycle and evaluated whether and how the ultradian rhythm in HRV interacts with the menstrual cycle. Based on the results of the study by Grant et al. [36], we hypothesized that we would observe a significant HF-HRV decrease in the periovulatory subphase (i.e., LH surge), but we did not have a hypothesis regarding which of the remaining five subphases this periovulatory trough would be significantly different from. We next examined whether progesterone mediates the hypothesized decrease in the periovulatory HF-HRV. Because progesterone begins to increase in the periovulatory subphase, and the HF-HRV is hypothesized to decrease, so we hypothesized an inverse relationship between HF-HRV and serum progesterone.

## 2. Materials and Methods

### 2.1. Study Sample and Study Design

The design of the Premenstrual Hormonal and Affective State Evaluation (PHASE) project is described in detail in [37]. In summary, reproductive-age women kept a daily record of their symptoms using the Daily Record of Severity of Problems (DRSP) [38] across two to three menstrual cycles. In the last menstrual cycle of the study, among other study measures, the participants completed HRV and blood sample collection visits at 8 different times during the menstrual cycle. PHASE-enrolled reproductive-age women with PMDD and healthy controls who were free of other current psychiatric conditions, could provide a drug-free urine sample, were non-smokers and did not take any prescription medications, including hormonal forms of birth control [37].

### 2.2. Study Procedures

Women participating in PHASE conducted ovulation testing using Clearblue urine tests, which was implemented to inform the scheduling of the study visits in the last menstrual cycle, not for fertility purposes. In the last menstrual cycle, study participants attended blood draw visits as described in detail in [37].

They completed the Daily Record of Severity of Problems [38], which is used to track menstrual-related symptomatology across the menstrual cycle. Although the present study was not powered to detect a subphase–diagnosis interaction, we assigned the diagnosis to repeat the analysis in healthy participants only, ensuring that the hypothesized periovulatory decrease in HRV is observed when restricted to the healthy participants. In accordance with DSM-5 criteria, PMDD diagnosis in the proposed study was assessed prospectively and was defined as a 30% or greater increase in symptoms [39].

### 2.3. Study Measures

High-Frequency Heart Rate Variability. HF-HRV corresponds to the heart rate variations related to the respiratory cycle, with a frequency band of 0.15 to 0.40 Hz, and reflects the parasympathetic nervous system’s influence on the heart [40]. Resting HF-HRV was collected for 5 min during morning visits while the participant was in a sitting position using the Zephyr^TM^ BioHarness 3.0 (Zephyr Technology Corporation, Annapolis, MD, USA), a continuous heart rate monitor. The analysis segment was defined as a 4 min period from the middle of the recording and 120 s out each way. Preprocessing of HF-HRV is described in detail in Hamidovic et al. [39].

Luteinizing Hormone. ARUP Laboratories, a CLIA-certified laboratory, performed serum LH analyses using a quantitative electrochemiluminescent immunoassay [41,42]. The reported limit of detection was 0.3 mIU/mL, and the lower limit of quantification was 1 mIU/mL.

Progesterone. Mass Spectrometry Core in the Research Resources Center at the University of Illinois at Chicago, as described in [43].

The Beck Depression Inventory (BDI). The BDI [44] is a 21-item self-report rating inventory that measures characteristic attitudes and symptoms of depression. Study participants completed the BDI to ensure that our protocol for screening out participants with current depression was efficient.

### 2.4. Data Analysis

We first realigned the study data as described in detail in [17]. If LH surges were not captured and luteal progesterone levels did not reach 5 ng/mL or higher, we considered the cycles anovulatory and did not analyze data collected from them. We then constructed the unconditional model (model 0) with individuals (IDs) nested within themselves and HF-HRV as the predictor using the linear mixed-effects model. We obtained model 0 statistics and the intraclass correlation coefficient [45] using the *lmer* function in R. We next added the subphase as a fixed slope (model 1) in order to evaluate how HF-HRV varies across the menstrual cycle for all 6 subphases. In this model, IDs were treated as a random effect. We could not complete higher level (subphase) nested in IDs because the number of observations was smaller than the number of random effects. We first tested the significance of model 1 using the *anova* function in R and compared model 0 and model 1 also using *anova*. Following this step, we collected the fixed and random effect statistics of model 1 using the *summary* function and re-calculated the ICC. We completed pairwise comparisons using the *lsmeans* function, corrected the resulting *p* value using the Tukey method for comparing a family of 6 estimates, and assessed the distribution of the residuals using the *qqnorm* function. To ensure a similar pattern in only healthy individuals, we repeated the same analysis steps and reported their results in the Appendix A. Given the limited power, we evaluated the most significant contrast from model 1 (HF-HRV at early follicular vs. periovulatory subphase) and constructed the final model (model 2), in which 2 subphases (early follicular and periovulatory) and the progesterone values were modeled as main effects, along with their interactive effects. Hence, subphase–progesterone was included as fixed effects, and ID was included as random effect.

## 3. Results

### 3.1. Study Participants and Menstrual Cycle Characteristics

Thirty-three women completed the study, out of which five did not have an ovulatory cycle. As our analytical approach controls for ovulation, we completed the analysis on the remaining twenty-eight women. Table 1 lists their demographic, anthropomorphic and psychological characteristics.

LH surges were captured in 22 out of the 28 (~78%) participants. This rate is similar to the findings of Mumford et al. [17], which validated the study methodology that we implemented in the present study as well as in our previous publication [37]. The mean and standard deviation periovulatory serum luteinizing hormone values in the 22 participants were 35.63 (14.26) IU/L. Values of progesterone for the 28 participants across the menstrual cycle are depicted in Figure 1.

### 3.2. High Frequency-Heart Rate Variability

The ICC of the unconditional means null model (model 0) showed that, of the total variance in HF-HRV, 68.86% was attributable to between-person variation. Including the subphase as a fixed effect term in the next model (model 1) indicated a significant effect of the subphase (*p* = 0.001831) and resulted in the following change from the intercept (β= 6.5380) (i.e., relative to the early follicular subphase, coded as 0): (1) mid-follicular (β = −0.4203, *p* = 0.0605); (2) periovulatory (β = −0.9302; *p* = 3.6 × 10^−6^); (3) early luteal (β = −0.2347, *p* = 0.2989); (4) mid-luteal (β = −0.4969, *p* = 0.0256); and (5) late luteal (β = −0.5709, *p* = 0.0201). The analysis of variance comparison showed a high statistical significance between models 0 and 1 (*p* < 2 × 10^−16^). The quantile plot (Appendix A) did not raise significant concerns regarding the normality of the weighted residuals. The ICC for model 1 was 72.14%. Pairwise comparisons of all the timepoints in model 1, with the Tukey method for comparing a family of six subphase estimates, showed statistically significant differences in HF-HRV between the early follicular and periovulatory subphases (β = 0.9302; *p* = 0.0005) and between the periovulatory and early luteal subphases (β = −0.6955; *p* = 0.0367). The full results of pairwise comparisons are presented in Table 2 and depicted in Figure 2. Restricting the analysis to healthy participants also only showed a statistically significant decrease in the periovulatory subphase of the menstrual cycle relative to the early follicular subphase. These results are presented in the Appendix A.

### 3.3. Progesterone by High Frequency-Heart Rate Variability Interaction

Model 2, which included main and interactive fixed effects of the subphase (early follicular and periovulatory), and which included individuals as random effects, demonstrated the following fixed effects findings: (1) progesterone (β = 1.0286; *p* = 0.0106); (2) subphase (periovulatory vs. early follicular) (β = −0.6613; *p* = 0.0392); and (3) subphase: progesterone interaction (periovulatory: progesterone vs. early follicular: progesterone) (β = −0.8487; *p* = 0.0350). As shown in Figure 3, the relationship between the predictor (progesterone) and the outcome (HF-HRV) was significantly different for the early follicular vs. periovulatory subphase.

## 4. Discussion

In line with our hypothesis, the results of the present study show that HF-HRV troughs occur in the periovulatory subphase of the menstrual cycle. By the start of the early luteal subphase, HF-HRV returns to baseline, reflected by a statistically different increase from the periovulatory to the early luteal subphase. In contrast to our second hypothesis, the present study shows that the positive correlation between HF-HRV and progesterone is no longer present in the periovulatory subphase. Hence, some other process, yet to be identified, is driving the marked periovulatory HF-HRV decrease.

The results of our study are not in agreement with any of the studies to date discussed in detail in the Introduction, with the exception of a recent study by Grant et al. [36], in which daily urinary samples from 16 premenopausal women were collected for measurements of estradiol, α-Pregnanediol and β-Pregnanediol. The women also wore a finger sensor during sleep, from which the ultradian power of HRV (RMSSD) was calculated. It showed an inflection point with a mean of 5.82 (±1.53) nights prior to LH surge onset, a subsequent peak with a mean of 2.58 (±1.59) nights prior to the surge onset and a trough with a mean of 2.11 (±1.27) days after surge onset (χ^2^ = 4.91, *p* = 0.034). Our study most likely reflects the trough identified by Grant et al. [36], as the mean and standard deviation of periovulatory serum luteinizing hormone values were 35.63 (14.26) IU/L. Hence, the present study extends this area of research by examining how additional subphases compare to the periovulatory HRV trough, thereby mapping changes in HRV across the entire menstrual cycle.

In the present study, we showed that the positive correlation between progesterone and HRV from the early follicular subphase (i.e., when HF-HRV) is high and is no longer present when the trough is reached in the periovulatory subphase. Thus, processes other than increasing levels of progesterone in the periovulatory subphase drive a drop in HF-HRV. Clearly, the mechanism of periovulatory troughs is an area of further research, as well as of clinical significance, if any, of the observed effect. Several studies presented in Thayer and Lane [46] indicate that there is an association between reduced CVA and an increased risk of diabetes [47], as well as hypertension [48]. As a significant independent predictor of mortality in individuals that have experienced myocardial ischemia, CVA has been suggested as a CAD risk factor [46,49]. Regarding the findings of the present study, whether the short-lived and cyclical periovulatory trough is a predisposition for ischemic or other acute events is presently unknown.

A limitation of our study is the loss of power due to the COVID-19 outbreak. Hence, we were unable detect a potential diagnosis (PMDD) according to the subphase or progesterone according to subphase interactions. To address the possible confounding of including PMDD participants in the total sample, we repeated the analysis in the healthy participants only, and, as in the full sample, we uncovered a decrease in the periovulatory relative to the early luteal subphase. Our approach would have been strengthened if we confirmed ovulation using ultrasonography; nonetheless, we utilized serum, not urinary, LH levels to realign the study data, which is a more valid approach [32]. Furthermore, given the limited power, we used the results of our analysis (model 1)—contrasting HRV across the entire menstrual cycle in a pairwise comparison—to select the most significant contrast and examine a potential interaction between that contrast and serum progesterone. We focused on progesterone because two prior studies (both studies described in Schmalenberger et al. [31]) examined the main effect of salivary progesterone and estradiol across the entire menstrual cycle on HRV, showing, in both studies, that salivary progesterone, but not estradiol, is negatively associated with HF-HRV. Salivary progesterone in the study was analyzed using immunoassay technology; however, relative to the mass spectrometry gold standard for the measurement of progesterone, immunoassay analysis of salivary progesterone may produce skewed results [50]. The present study strengthens this area of research as progesterone was analyzed using ultraperformance liquid chromatography tandem mass spectrometry.

This study further demonstrates the value of understanding relationships between infradian rhythms and ultradian rhythms, in the present study characterized as an interaction between the menstrual cycle and the HF-HRV. In addition to reasons related to tracking fertility, as described in Grant et al. [36], HF-HRV provides insight into autonomic flexibility and corresponding emotional regulation tools available to individuals [14,51]. Changes in HF-HRV could potentially underlie some of the deficits in cognitive and emotional functioning seen during ovulation [52]. This possibility is particularly intriguing because HF-HRV is amenable to intervention [53,54]. Additional studies directly measuring the cognitive and affective corollaries of HF-HRV and using experimental manipulations of HF-HRV will be required to test the role of periovulatory dips in HF-HRV in cognitive and affective changes over the menstrual cycle. However, the current study is critical to unlocking these possibilities, as the more accurate method of cycle determination used here revealed ultradian changes in HF-HRV relative to the entire menstrual cycle that have not been seen in prior studies. This study reveals yet another inflection point where rapid changes in physiological functioning take place, this time in HF-HRV, that may be productively studied now that it has been identified by more fine-grained cycle tracking.

## 5. Conclusions

The results of the present study show a significant periovulatory trough at the time of the short-lived LH surge relative to all other time periods of the menstrual cycle. Numerous short-lasting (several per hour) reproductive biological rhythms coordinate with the periovulatory subphase, including ultradian frequency shifts in steroid hormones. Non-reproductive measures, such as changes in HF-HRV, reflect the activity of the reproductive system in anticipation of ovulation. Whether ultradian rhythms reflect reproductive changes in the periovulatory subphase, whereas circadian rhythms are more phase-specific, is an area of future research, as is the question of whether the periovulatory trough in HF-HRV has any clinically meaningful implications or whether it simply reflects the flexible state of the female organism to balance numerous rhythms simultaneously.

## Figures and Tables

**Figure 1 biology-12-00785-f001:**
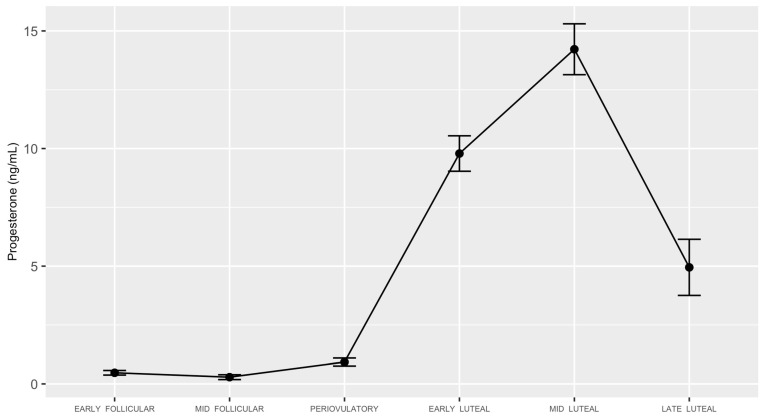
Serum progesterone concentrations of study participants with ovulatory cycles.

**Figure 2 biology-12-00785-f002:**
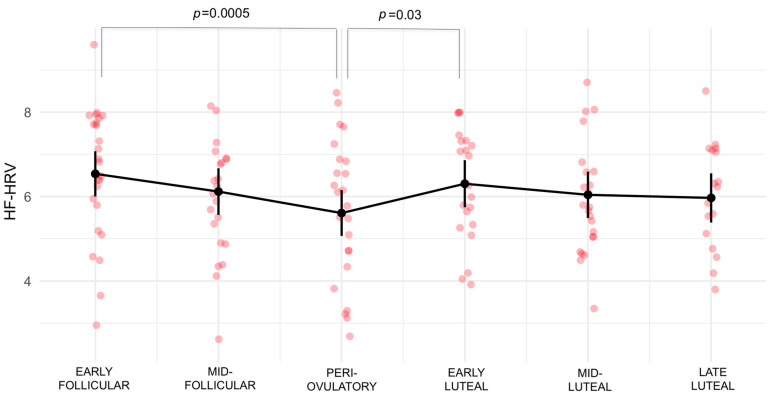
HF-HRV estimates and standard errors of the prediction model (model 1) according to the six subphases of the menstrual cycle. Adjusted *p* values from pairwise comparisons revealed statistically significant differences between early follicular and periovulatory subphases (*p* ≤ 0.001) and between periovulatory and early luteal (*p* ≤ 0.05) subphases.

**Figure 3 biology-12-00785-f003:**
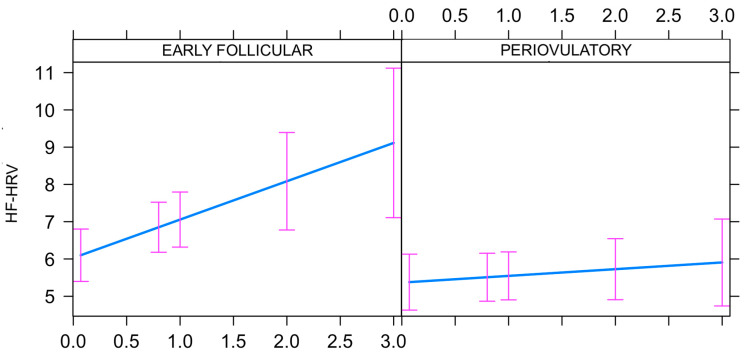
The relationship between the predictor (progesterone) and the outcome (HF-HRV) was identified to be significantly different for the early follicular vs. periovulatory subphase (*p* ≤ 0.05). It is presented as estimates and standard errors of the prediction model. Main effects of progesterone and subphase were also significant (*p* ≤ 0.05).

**Table 1 biology-12-00785-t001:** Demographic and anthropomorphic characteristics of study participants with ovulatory cycles (*n* = 28).

Category	Mean (SD) or N
AGE	
	26.53 (5.01)
RACE	
White	10
African American	7
American Indian/Alaska Native	1
Asian	5
Native Hawaiian or Other Pacific Islander	0
More than 1 race	1
Unknown/Do not want to specify	4
ETHNICITY	
Hispanic	9
Non-Hispanic	17
Unknown/Do not want to specify	2
STUDENT STATUS	
Yes	15
No	13
MARITAL STATUS	
Single/Never married	25
Married	2
Divorced	1
INCOME	
Less than USD 20,000	13
USD 20,000–34,999	5
USD 35,000–49,999	4
USD 50,000–74,999	3
USD 75,000 or more	3
MENARCHE AGE	
	11.89 (1.44)
BMI *	
	25.55 (4.75)
BDI **	
	2.60 (3.10)

* BMI = Body Mass Index; ** BDI = Beck’s Depression Inventory.

**Table 2 biology-12-00785-t002:** Pairwise comparisons of HF-HRV Across the six subphases of the menstrual cycle.

Contrast	Estimate	SE	df	t-Ratio	*p*-Value
Early follicular–Mid-follicular	0.4203	0.221	101	1.898	0.4093
Early follicular–Periovulatory	0.9302	0.215	100	4.324	0.0005 ***
Early follicular–Early luteal	0.2347	0.225	101	1.044	0.9019
Early follicular–Mid-luteal	0.4969	0.219	101	2.265	0.2182
Early follicular–Late luteal	0.5709	0.242	101	2.361	0.18
Mid-follicular–Periovulatory	0.5098	0.228	101	2.236	0.231
Mid-follicular–Early luteal	−0.1857	0.234	101	−0.794	0.9679
Mid-follicular–Mid-luteal	0.0765	0.23	101	0.333	0.9994
Mid-follicular–Late luteal	0.1506	0.254	102	0.592	0.9914
Periovulatory–Early luteal	−0.6955	0.23	101	−3.02	0.0367 *
Periovulatory–Mid-luteal	−0.4333	0.224	101	−1.93	0.3901
Periovulatory–Late luteal	−0.3593	0.247	101	−1.455	0.6931
Early luteal–Mid-luteal	0.2622	0.235	101	1.118	0.8728
Early luteal–Late luteal	0.3362	0.258	102	1.304	0.7824
Mid-luteal–Late luteal	0.074	0.249	101	0.298	0.9997

* *p* ≤ 0.05; *** *p* ≤ 0.001.

## Data Availability

The data presented in this study are available upon request from the corresponding author. The data are not publicly available to preserve the scientific integrity of the research methodology.

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
