# Peer review of "Periovulatory Subphase of the Menstrual Cycle Is Marked by a Significant Decrease in Heart Rate Variability"

_biology, 2023, doi:10.3390/biology12060785_

Round 1

Reviewer 1 Report

This paper describes the variations of heart rate variability in relation to the periovulatory subphases of the menstrual cycle in healthy women.

Results show a decrease in parasympathetic activity prior to ovulation.

The paper is interesting and well written, however I have some concerns.

Introduction should cite more reference paper investigating autonomic control/HRV according to the menstrual cycle. Citing a single systematic review is not enough.

Line 210: better define HF-HRV and the relative physiological meaning and frequency band.

A paragrah describing statistical analysis would be ppreciated, e.g to describe differences in table

Using only HF does not represent the overall HRV.Please give other indices as mean RR interval, total variance or SDNN, sympathovagal balance as the ratio of the power in LF (low frequency, 0.04-0.15) on that in HF band, LF/HF.

The relative methods should be described in methods section.

Discussion: please better describe the physiological meaning of findings with also more citations about HRV and its pathophysiological correlates.

Line 427: why is COVID-19 a limitation? Please explain.

Author Response

This paper describes the variations of heart rate variability in relation to the periovulatory subphases of the menstrual cycle in healthy women. Results show a decrease in parasympathetic activity prior to ovulation. The paper is interesting and well written, however I have some concerns.

COMMENT 1: Introduction should cite more reference paper investigating autonomic control/HRV according to the menstrual cycle. Citing a single systematic review is not enough.

RESPONSE 1: We thank the reviewer for all the comments. We added additional reference papers (lines 201-213) in the Introduction that have investigated HRV according to the menstrual cycle.

COMMENT 2: Line 210: better define HF-HRV and the relative physiological meaning and frequency band.

RESPONSE 2: We have defined HF-HRV and provided the physiological meaning as well as the frequency band.

COMMENT 3: A paragraph describing statistical analysis would be appreciated, e.g to describe differences in table.

RESPONSE 3: The methods used to perform data analyses have been described in section 2.5. Results of data analyses have been presented in sections 3.2 and 3.3. Statistically significant differences in comparisons of HF-HRV during the early follicular and periovulatory subphases as well as the periovulatory and early luteal subphases. No other significant differences were observed in subphase pairwise comparisons presented in Table 2.

COMMENT 4: Using only HF does not represent the overall HRV.Please give other indices as mean RR interval, total variance or SDNN, sympathovagal balance as the ratio of the power in LF (low frequency, 0.04-0.15) on that in HF band, LF/HF. The relative methods should be described in methods section.

RESPONSE 4:  We acknowledge the reviewer’s point that only high-frequency HRV (HF-HRV) is represented in the paper, not other aspects of HRV such as low-frequency (LF-HRV), or ultra-low frequency (ULF-HRV).  We regret using just heart rate variability (HRV) as our terminology in the paper, as this created confusion about our focus.  Our interest is in the functioning of the parasympathetic nervous system across the menstrual cycle, as represented by high-frequency HRV (HF-HRV).  We now use the term high-frequency HRV (HF-HRV) throughout instead of just HRV.  Regarding examining LF-HRV and contributions of sympathetic/parasympathetic balance, we did consider including LF/HRV and LF/HF balance but elected not to because connection between LF-HRV and sympathetic activity is controversial in the literature, and LF-HRV and LF/HF metrics have been repeatedly criticized as potentially misleading (Quintana et al., 2016, Billman, 2013). Thus, we do not believe including LF-HRV would add to the report. Regarding HF-HRV, it is also the case that there are many ways of quantifying HF-HRV (over 70 published metrics per Quintana et al., 2016), including the types of time-domain methods noted by the reviewer, frequency domain methods, etc. Generally the results of these methods are highly correlated, but that does not mean they are perfectly interchangeable. We carefully selected the moving polynomial method we used for quantifying HF-HRV for several reasons: 1. It more strongly reflects contributions of the parasympathetic nervous system, our primary interest, compared to other methods like SDNN (Zahn et al., 2016). 2. It shows superior statistical properties to standard time-domain and frequency-domain methods (Lewis et al., 2012) 3. It does not require correction for respiration frequency, simplifying data collection (Lewis et al., 2012, Denver et al., 2007). Given these reasons, we feel that inclusion of alternate metrics of HF-HRV that do not as accurately index the parasympathetic processes of interest would not improve the paper. This is now discussed in Introduction (lines 86-106).

COMMENT 5: Discussion: please better describe the physiological meaning of findings with also more citations about HRV and its pathophysiological correlates.

RESPONSE 5: Additional details and citations have been included in the discussion to support that low HRV is indicative of poor health due to a greater risk of diabetes, hypertension, cardiovascular disease, and emotion dysregulation (lines 1272-1278).

COMMENT 6: Line 427: why is COVID-19 a limitation? Please explain.

RESPONSE 6: We revised this section, which now reads: “A limitation of our study is the COVID-19 outbreak during which we were not able to collect study data, resulting in the lack of power to detect potential diagnosis (PMDD) by subphase, or progesterone by subphase interactions.”

Reviewer 2 Report

The paper has an interesting and important theme for the field of study, however some corrections are necessary.

Simple Summary - " Heart rate variability measures differences in time between heart beats, and reflects the activity of the parasympathetic nervous system."

It is important to make it clear that the HRV does not only reflect the parasympathetic nervous system, but all the activity of the autonomic nervous system, with the parasympathetic nervous system being part of it.

"As such, low heart rate variability reflects low parasympathetic tone."

Same thing as the previous one.

Abstract: "It is unknown how HRV varies across the MC"

MC acronym not previously defined

Methods: RMSSD is cited but not shown in the results.

Discussion: Well written and clear text

Conclusion: Well defined and related to the objective

Author Response

The paper has an interesting and important theme for the field of study, however some corrections are necessary.

COMMENT 1: Simple Summary - " Heart rate variability measures differences in time between heart beats, and reflects the activity of the parasympathetic nervous system." It is important to make it clear that the HRV does not only reflect the parasympathetic nervous system, but all the activity of the autonomic nervous system, with the parasympathetic nervous system being part of it. "As such, low heart rate variability reflects low parasympathetic tone." Same thing as the previous one.

RESPONSE 1: We thank the reviewer for pointing this important aspect. We have significantly revised the simple summary, which now reads:

Simple Summary: If measured in the high-frequency component (HF), heart rate variability (HRV) reflects the tonic, inhibitory control by the parasympathetic nervous system (PNS) to decelerate the heart rhythm. Several studies examined changes in HF-HRV variability across the menstrual cycle; however, they did not implement the latest technological and statistical approaches. The evaluation of HF-HRV across the menstrual cycle is important because low HF-HRV variability is observed in numerous conditions, including, heart disease, and depression. Moreover, if a change in HF-HRV is observed around ovulation, it may be used to time optimal conception. This study found that HF-HRV significantly decreases around ovulation relative to all other times of the menstrual cycle. This finding enhances our understanding of human biology. Future studies may examine whether the decrease in HF-HRV is a normal process that may be utilized in the future to enhance fertility rates, or it represents a biological process which may predispose some women to disease development.

COMMENT 2: Abstract: "It is unknown how HRV varies across the MC. MC acronym not previously defined.

RESPONSE 2: We have clarified this acronym in the abstract.

COMMENT 3: Methods: RMSSD is cited but not shown in the results.

RESPONSE 3: We have removed the discussion on RMSSD to maintain the focus of the manuscript on HF-HRV.

COMMENT 4: Discussion: Well written and clear text.

RESPONSE 4: We thank the reviewer.

COMMENT 5: Conclusion: Well defined and related to the objective.

RESPONSE 5: We thank the reviewer.

Reviewer 3 Report

With great interest, I read the manuscript on an important clinical issue that significantly impacts the clinical problems of many patients. The matter is certainly of the utmost importance, affecting large groups of women around the world. Authors show interesting perspective underlying important but often omitted aspects of reproductive health. Nevertheless, authors did not shy away from minor shortcomings and deficiencies, which I will present in points:

- v 46-48 - The sentence is misleading. There has been a confusion here between the rhythm of daily and monthly temperatures. In fact, in humans there is a diurnal rhythm of temperature - related to activity. There is also a basal body temperature (BBT), the rhythm of which changes (only in menstruating women) during the menstrual cycle - associated with ovulation/progesterone secretion (PMID: 36833150). Basal body temperature can be measured at night or upon awakening. Daytime temperature has no such relationship, as it is largely associated with physical activity. These phenomena should be accurately described, as the statement is erroneous in the current situation.

- v 70-73 - That the luteal phase has a fixed length and does not change between women is an oversimplification as far as scientific research is concerned. It is known that many women have luteal phase deficiency - manifested mainly by a shortened luteal phase. Please refer to the ASRM guidelines (PMID: 22819186).

- A major limitation that has not been mentioned is the lack of correlation of presumed ovulation with methods that accurately assess ovulation. Hormonal measurements are known to correlate only to a certain extent with the actual day of ovulation. The method that is capable of confirming an ovulatory event, and demonstrating subtle ovulatory abnormalities, is only ultrasonography. This method, although available and non-invasive, has not been used at all. I think that if the prediction of ovulation was more accurate then the relationship with HRV would also be clearer. The inaccuracy of biochemical tests is described in up to 7% of regular cycles PMID: 15533341. 

Author Response

With great interest, I read the manuscript on an important clinical issue that significantly impacts the clinical problems of many patients. The matter is certainly of the utmost importance, affecting large groups of women around the world. Authors show interesting perspective underlying important but often omitted aspects of reproductive health. Nevertheless, authors did not shy away from minor shortcomings and deficiencies, which I will present in points:

COMMENT 1: - v 46-48 - The sentence is misleading. There has been a confusion here between the rhythm of daily and monthly temperatures. In fact, in humans there is a diurnal rhythm of temperature - related to activity. There is also a basal body temperature (BBT), the rhythm of which changes (only in menstruating women) during the menstrual cycle - associated with ovulation/progesterone secretion (PMID: 36833150). Basal body temperature can be measured at night or upon awakening. Daytime temperature has no such relationship, as it is largely associated with physical activity. These phenomena should be accurately described, as the statement is erroneous in the current situation.

RESPONSE 1: We thank the reviewer for this comment. In this section, we clarified that the change in body temperature as a function of menstrual cycle phase refers to basal body temperature, and removed “circadian”  from the paragraph. The section now reads:” Physiological rhythms are distinct time-related processes that are essential for the survival of the organism. They span several orders of magnitude; from microsecond to seasonal events [1, 2]. Lasting more than 24 hours, the ovulatory menstrual cycle is an infradian rhythm [3] during which basal body temperature, for example, increases by 0.4 °C on average compared to the follicular phase [4-6]. This and other rhythms synchronize cellular activities to enhance the stability of the organism, or alternatively, fail to do so, causing perturbations and disease progression [7].”

COMMENT 2: - v 70-73 - That the luteal phase has a fixed length and does not change between women is an oversimplification as far as scientific research is concerned. It is known that many women have luteal phase deficiency - manifested mainly by a shortened luteal phase. Please refer to the ASRM guidelines (PMID: 22819186).

RESPONSE 2: We agree with the reviewer, and do not advocate for the backward counting method. In fact, we emphasize in the manuscript that the Biocycle method, which we implemented in the study, outperformed the backward counting method (along with the fixed and mid-point cycle methods). Moreover, it is emphasized that prior works assessing HRV changes across the menstrual cycle were likely flawed precisely due to implementation of these three, non-validated methods.

COMMENT 3: - A major limitation that has not been mentioned is the lack of correlation of presumed ovulation with methods that accurately assess ovulation. Hormonal measurements are known to correlate only to a certain extent with the actual day of ovulation. The method that is capable of confirming an ovulatory event, and demonstrating subtle ovulatory abnormalities, is only ultrasonography. This method, although available and non-invasive, has not been used at all. I think that if the prediction of ovulation was more accurate then the relationship with HRV would also be clearer. The inaccuracy of biochemical tests is described in up to 7% of regular cycles PMID: 15533341. 

RESPONSE 3: We thank the reviewer for pointing this out, and we have added this in the limitation section, which now reads: “Our approach would have been strengthened if we confirmed ovulation using ultrasonography; nonetheless, we utilized serum, not urinary, LH levels to realign the study data, which is a more valid approach [32].”

Round 2

Reviewer 1 Report

Authors have addressed this reviewer's concerns and the paper has been improved.
I only suggest to correct two minor typos: 

- line 141-142: frequency (...) and time (...) -> frequency and time domain markers

- line 368 (2030) -> (2023)

Author Response

We thank the reviewer for the careful review of our manuscript. We have updated the next version (submitted here) with the two corrections. 
